# Mulberry *MnGolS2* Mediates Resistance to *Botrytis cinerea* on Transgenic Plants

**DOI:** 10.3390/genes14101912

**Published:** 2023-10-06

**Authors:** Donghao Wang, Zixuan Liu, Yue Qin, Shihao Zhang, Lulu Yang, Qiqi Shang, Xianling Ji, Youchao Xin, Xiaodong Li

**Affiliations:** 1Guangxi Key Laboratory of Sericulture Ecology and Applied Intelligent Technology, Hechi University, Hechi 546300, China; qinyue1161@163.com (Y.Q.); 2021660014@hcnu.edu.cn (S.Z.); 2College of Forestry, Shandong Agricultural University, Tai’an 271018, China; wangdh1204@163.com (D.W.); 18863329786@163.com (Z.L.); m15582668885@163.com (L.Y.); 17686268716@163.com (Q.S.); xlji@sdau.edu.cn (X.J.); 3Guangxi Collaborative Innovation Center of Modern Sericulture Silk, School of Chemistry and Bioengineering, Hechi University, Hechi 546300, China

**Keywords:** galactinol synthase, *B. cinerea*, mulberry, *MnGolS2*

## Abstract

Galactitol synthetase (GolS) as a key enzyme in the raffinose family oligosaccharides (RFOs) biosynthesis pathway, which is closely related to stress. At present, there are few studies on GolS in biological stress. The expression of *MnGolS2* gene in mulberry was increased under *Botrytis cinerea* infection. The *MnGolS2* gene was cloned and ectopically expressed in *Arabidopsis*. The content of MDA in leaves of transgenic plants was decreased and the content of CAT was increased after inoculation with *B. cinerea*. In this study, the role of *MnGolS2* in biotic stress was demonstrated for the first time. In addition, it was found that *MnGolS2* may increase the resistance of *B. cinerea* by interacting with other resistance genes. This study offers a crucial foundation for further research into the role of the GolS2 gene.

## 1. Introduction

Raffinose family oligosaccharides (RFOs) exist widely in plants. RFOs is a special functional oligosaccharide in plants and its content in most plant seeds is second only to sucrose. Studies have shown that RFOs play an important role in seed dehydration tolerance, seed storage and seed vitality, as well as abiotic and biological stresses such as temperature, salt, drought, oxidation and pathogenic bacteria [1,2,3,4,5,6]. Galactinol synthase, whose activity determines the accumulation level of some RFOs, is considered to be a key regulatory enzyme of the RFO pathway. It has been found that there are four GolS genes in the cucumber genome, and the tissue specificity of the four GolS genes was studied via the GUS reporter gene method, and GolS1 gene could be specifically expressed in the phloem of the cucumber. This gene can not only increase its expression under stress such as low temperature but also promote the loading of assimilates in leaf small vascular bundle phloem under stress. Through the dual effects of the GolS1 genes low temperature response and phloem loading, the output efficiency of assimilates in leaves was improved, photosynthesis of leaves was enhanced, and the overall growth performance of cucumber plants under stress was improved [7,8]. Galactinol synthase plays a significant part in carbohydrate metabolism, which has led to intensive research on the GolS genes, which are involved in many plant species under varied stress circumstances. *TaGolS3* is mainly expressed in roots. Moreover, it was up-regulated under abiotic stress such as ZnCl2, CuCl2, low temperature and NaCl. In addition, compared with wild-type plants, *TaGolS3* overexpression lines showed higher ROS scavenging gene expression, antioxidant enzyme activity, proline content and lower malondialdehyde (MDA) content. The tolerance of tobacco to drought and salt stress was significantly improved under conditions of CsGolS6 overexpression, promoted mesophyll cell expansion, photosynthesis and plant growth [9,10]. Some studies have shown that the *CaGolS* gene is induced in order to express in small-grained coffee under drought and salt stress conditions [11]. Under the MV treatment, *GolS1,2,3,4* and *8* were induced to express in *Arabidopsis*. Plant tolerance to high salt and high osmotic stress can be achieved by expressing *TsGolS2* [12]. In transgenic *Arabidopsis*, overexpression of *AtGolS1* and *AtGolS2* increased drought resistance and clearance of reactive oxygen species, along with increased accumulation of galactinol and raffinose [13]. Abiotic stress strongly induced several key gene members. qRT-PCR was used to further verify the expression level [14]. Changes in the transcription levels of various defense genes were observed in Marmarine-treated cucumbers, including cucumber-pathogen-induced gene 4 and galactinol synthase [15].

Mulberry (*Morus* L.) is widely planted because of its good economic and ecological value [16,17,18,19,20]. Mulberry is rich in genetic resources. It is not only the basis of global sericulture, but also plays an important role in human health and ecological environment protection because of its rich secondary metabolites. Compared with the chemosynthetic drugs, the mulberry Chinese patent medicine with 1-deoxynojirimycin as the main active ingredient in the treatment of type 2 diabetes showed the same efficacy as the mainstream combined drugs and alleviated the side effects caused by the combined drugs. Mulberry is the plant with the highest content of DNJ [21,22,23]. *B. cinerea* is a disease caused by fungal infection. This fungus can erode many parts of plants such as leaves, flowers, fruits and stems. *B. cinerea* is mainly caused by sclerotia or mycelia and conidial stems left in the soil over summer or winter with diseased residues. When environmental conditions such as temperature and humidity are suitable, the sclerotia germinates to produce a large number of hyphae, and conidia are produced on the hyphae. Conidia and hyphae are transmitted by air, running water and people’s production activities, which can cause diseases in tomatoes, roses and other plants. At the same time, *B. cinerea* can infect mulberry plants and cause serious economic loss [24,25,26,27]. The accumulation of galactinol in plants in response to pathogen infection, it is involved in the plant resistance system to pathogens [28,29]. As an important step in the synthesis of galactinol, galactinol synthetase is related to plant resistance to pathogens. GolS genes functions in biological stress were rarely reported. It is important to further study the function of GolS gene to reveal the molecular mechanism between plants and *B. cinerea*.

In this study, *MnGolS2* was used to reveal its role in plant response to *B. cinerea*. After *B. cinerea* infection, the expression level of *MnGolS2* in mulberry dramatically increased. To study the resistance function of *MnGolS2*, it was heterologous expressed in *Arabidopsis*. Through a series of studies on transgenic *Arabidopsis*, it has been suggested that the pathogen defense involves *MnGolS2* in transgenic plants. The experimental results preliminarily revealed the disease resistance mechanism of *MnGolS2*, and offer a crucial foundation for further research into the role of the GolS2.

## 2. Materials and Methods

### 2.1. Plant Materials and Equipment

The variant of *Arabidopsis* used is the Columbia type. Mulberry (*Morus* L.) and *Arabidopsis thaliana* plants were cultured at 24 °C, with humidity levels of 50–60% and 16 h of light every day. In this study, we used CFX Connect fluorescence quantitative PCR instrument (Bio-Rad, Hercules, America), double beam ultraviolet visible spectrophotometer (Pgeneral, Beijing, China), and other instruments. The mature mulberry fruit served as the source for the MM1 isolate of *B. cinerea.* The culture temperature of *B. cinerea* was 28 °C.

### 2.2. Phylogenetic Tree of GolSs

To study the evolutionary relationships of GolS, we aligned the amino acid sequences of GolS proteins with other species and subsequently constructed a neighborhood joining phylogenetic tree of GolS using MEGA 11 [30]. One thousand replicates were used in the Bootstrap analysis.

### 2.3. Quantitative Real-Time PCR

Mulberry and *Arabidopsis* leaves were inoculated with *B. cinerea* and agar blocks, respectively. Using TRIzol reagent (Invitrogen, Carlsbad, CA, USA) separation of RNA in the samples and repeated three times. qRT-PCR was performed using QuantiNova ^TM^ SYBR Green PCR kit (Qiagen, Hilden, Germany) and Step OnePlus ^TM^ real-time PCR system (Applied Biosystems, Waltham, MA, USA). The internal control genes of mulberry and *Arabidopsis* were *MnActin* and *AtActin*, respectively. qRT-PCR was repeated three times. Appendix A provides specifics on qRT-PCR primers.

### 2.4. Construction of Expression Vector and Plant Transformation

The sequence of *MnGolS2* was cloned into the *Kpn*I and *EcoR*I restriction sites of the pLGNL vector under the control of the *CaMV35s* promoter to generate the *CaMV35s::MnGolS2* recombinant plasmid. *MnGolS2* were transferred to *Arabidopsis* via inflorescence impregnation. Transgenic plants can be obtained by screening the seeds of infected *Arabidopsis*, and then selfing until homozygous lines are obtained [31].

### 2.5. Analysis of Resistance of Transgenic Arabidopsis

The transgenic seeds were spread on 1/2 Murashige and Skoog agar medium. After 7 days of germination, the transgenic seedlings were transplanted into a nutrient soil pot and grown at 24 °C under a 16 h light/8 h dark photoperiod. Previously isolated and identified *B. cinerea* strains were cultured in Potato Dextrose Agar for 5–7 days at a temperature of 28 °C. *B. cinerea* was applied to the leaves of plants that were 21 days old. After 36 h, photographs were taken of the inoculated plants. According to the lesion area, the degree of disease was identified.

As controls, pLGNL-transgenic Arabidopsis plants were employed. The malondialdehyde assay kit (Solarbio, Beijing, China) was used to measure the amount of malondialdehyde (MDA) and the catalase (CAT) activity. The process is as follows. Weigh 0.1 g of the sample to be measured, add 1 mL of the extraction solution, ice bath homogenate, centrifuge at 8000× *g* 4 °C, the supernatant was taken to be tested, add 35 μL of the sample into 1 mL CAT test solution, mix it well for 5 s at room temperature to determine the initial light absorption value at 240 nm and the light absorption value after 1 min, and calculate the CAT activity according to the instructions. For the determination of MDA, it is necessary to mix the MDA detection working liquid, distilled water, sample, reagent 3 according to a certain proportion, hold them in 100 °C water for 60 min, cool them in an ice bath for 1000× *g*, centrifuge at room temperature for 10 min, take supernatant, measure the absorbance of 450 nm, 532 nm and 600 nm, respectively, and determine their content according to the instructions. The treatments were given in triplicate.

Histochemical staining was used to assess the superoxide radical (O_2_^−^) and hydrogen peroxide (H_2_O_2_) concentrations in leaves. 0.1% nitroblue tetrazole (NBT) containing 50 mM potassium phosphate buffer (pH = 7.8) was used for O_2_^−^ detection. Detection of H_2_O_2_, the 3,3′-diaminobenzidine (DAB) kit (Solarbio, Beijing, China) was used. *Arabidopsis* leaves were placed into the configured DAB and NBT dyeing solution for dyeing. After a period of time, the leaves were taken out for decolorization, and then photos were taken [32].

### 2.6. Statistical Analysis

Using SPSS 26.0 software, Student’s *t*-test and one-way ANOVA were applied to the data in this paper. These figures are given as mean standard deviation (SD).

## 3. Results

### 3.1. GolSs Bioinformatics Analysis

Multiple alignments of GolSs protein sequences from other plants obtained from NCBI were performed. MEGA11 was used for phylogenetic and molecular evolution analysis to explore the evolutionary relationship between different plants (Figure 1). Results showed that mulberry was closely related to *Cannabis sativa*, but far from *Rosa chinensis* and *Glycine max*.

### 3.2. B. cinerea-Induced MnGolS2 Expression

In order to verify whether *MnGolS2* plays a role in the infection of *B. cinerea*, agar block and agar blocks with *B. cinerea* mycelia were placed on mulberry leaves, and then sample RNA was extracted, respectively. The expression of *MnGolS2* was observed by qRT-PCR, and it was found that the *MnGolS2* gene increased significantly after 3 days of inoculation with *B. cinerea* (Figure 2).

### 3.3. Ectopic Expression of MnGolS2 in Arabidopsis

In order to further verify the resistance function of *MnGolS2* gene, we performed heterologous expression of *MnGolS2* gene. Under the control of *Cauliflower mosaic virus* 35S promoter, *MnGolS2* cDNA was introduced into *Arabidopsis* to obtain T3 transgenic plants. *MnGolS2* gene was verified by qRT-PCR (Figure 3a), and GUS staining also proved the expression of the *MnGolS2* gene. After GUS staining, the transgenic plants appeared blue (Figure 3b).

### 3.4. Positive Regulation of MnGolS2 for Resistance

To determine the resistance of *MnGolS2* transgenic *Arabidopsis*, the leaves containing empty vector and *MnGolS2* transgenic *Arabidopsis* were inoculated with *B. cinerea*. After 36 h of inoculation, the leaves of the empty vector control showed severe lesions, while *MnGolS2* overexpression lines showed only mild lesions (Figure 4a). The contents of hydrogen peroxide (H_2_O_2_) and superoxide anion (O^2−^) in leaves were detected by 3,3′-diaminobenzidine and nitroblue tetrazole staining. Phenotypically, compared with *MnGolS2* transgenic *Arabidopsis*, the empty vector transgenic *Arabidopsis* showed a larger amount of dark brown after 3,3′-diaminobenzidine staining, indicating that H_2_O_2_ accumulated and showed a larger amount of dark blue after nitroblue tetrazole staining, which was a marker of O^2−^ (Figure 4b). In addition, the quantitative results showed that the *MnGolS2* transgenic *Arabidopsis* had a certain inhibitory effect on the spread of *B. cinerea* (Figure 4c).

### 3.5. Biochemical Index Detection

The MDA concentration and CAT activity of *MnGolS2* transgenic *Arabidopsis* and the empty vector were measured to confirm the physiological alterations (Figure 5). No significant difference in MDA or CAT content between *MnGolS2* transgenic *Arabidopsis* and empty vector under normal circumstances. In the case of *B. cinerea* infection, *MnGolS2* transgenic plants had significantly higher CAT activity than empty vector transgenic plants (Figure 5a), but empty vector transgenic plants had higher MDA content (Figure 5b). These findings suggested that empty vector transgenic plants suffered more severe plasma membrane damage than *MnGolS2* transgenic plants. The enhancement of oxidative damage resistance can be achieved by transferring the *MnGolS2* gene.

### 3.6. Transgenic MnGolS2 Plants Enhance the Expression of Resistance-Related Genes

PR1 and β-1,3-glucanase 2 (BG2) are defense-related marker genes in plants. Showed that the expression levels of *AtPR1* and *AtBG2* in *MnGolS2* transgenic *Arabidopsis* were significantly higher than those in empty vector transgenic *Arabidopsis* under *B. cinerea* infection (Figure 6). In the absence of *B. cinerea* infection, there was no difference between the *MnGolS2* gene transgenic *Arabidopsis* and the empty vector transgenic *Arabidopsis*. These results suggest that the introduction of the *MnGolS2* gene into *Arabidopsis* can prevent the infection of *B. cinerea* by inducing the expression of resistance-related genes.

## 4. Discussion

Galactinol synthase (GolS) has been found in *Arabidopsis*, potato, cucumber and other species [33,34,35]. Studies have shown that GolS genes were isolated from the leaves of *Populus nigra* and the expression patterns of GolS genes in response to hypoxic acid (ABA), salinity, drought or cold stress were shown. *PnGolS2* gene was found and its transcription level was significantly increased in response to stress, but whether it responds to biological stress has not been reported [36]. Under drought, osmosis, salt stress and waterlogging stress, *SiGolS* and *SiRS* genes were significantly regulated, but cold stress had little effect [37]. It has been shown that *CsGolS4* was highly expressed in mesophyll cells and leaf vascular bundles. Overexpression of *CsGolS4* increased raffinose family oligosaccharide content and inhibited the production of reactive oxygen species, while *CsGolS4* RNAi transgenic plants had the opposite effect. Therefore, it is suggested that *CsGolS4*-mediated stress tolerance may be related to oligosaccharides content, and can be used as osmoprotectants to promote ROS clearance. *CaGolS* is mediated by protective cell membranes from ROS, which attack dehydration stress tolerance [38,39], but whether it responds to biotic stress has not been reported. Through experiments, it is further proven that this situation still exists under biological stress. In order to study the function of galactinol synthase, the galactinol synthase gene should be ectopically expressed in *Arabidopsis* and then inoculated with *B. cinerea* (Figure 4). Malondialdehyde (MDA) is the final decomposition product of membrane lipid peroxidation; its content can reflect the damage degree of plants subjected to stress. Much of the cell membrane and cell damage is caused by the accumulation of malondialdehyde. Through the MDA can understand the level of membrane lipid peroxidation, indirect measure of damage degree of membrane system and resistance of plants. Its content can be used as one of the indicators to evaluate the severity of stress on cells. Its main harm is to cause membrane lipid peroxidation, the biofilm structure was destroyed, mainly the plasma membrane of cells, damage the structure and function of the cell membrane, change the permeability of the membrane, thus affect the normal progress of physiological and biochemical reactions. As an antioxidant enzyme, CAT enzyme is mainly used to decompose hydrogen peroxide in cells into water and oxygen, thereby reducing the degree of oxidative stress in cells. DAB and NBT staining were used to find the ROS activity (Figure 4). The leaves of *MnGolS2* transgenic plants displayed less damage and ROS buildup than the leaves of empty plants, increasing their resilience, which was consistent with our previous research results.

The accumulation of galactinol and RFOs increased in a few plant seeds. In the process of seed development and germination, *CaGolS1* and *CaGolS2* were regulated by different organs, but showed similar subcellular localization. In addition, specific overexpression of *CaGolS1* and *CaGolS2* in seeds increased seed viability and longevity by limiting excess ROS and lipid peroxidation induced by aging, Raffinose is significantly accumulated in the chloroplasts of the antifreeze leaves of cabbage (*Brassica oleracea*) [40,41]. Plant species play a decisive role in the effect of stachyose on environmental stress. In *Arabidopsis*, after drought, high salinity, and cold treatment, large amounts of raffinose and galactinol accumulate, but no sugar accumulates. This suggests that raffinose and galactinol are associated with drought tolerance, high salinity and cold stress. Three stress-responsive GolS genes were identified in *Arabidopsis*. Drought and high salt stress can cause high expression of AtGolS1 and AtGolS2, while cold stress can cause *AtGolS3* expression. Stressinduced galactinol synthetase plays a key role in the accumulation of galactinol and lactose under abiotic stress, galactinol and raffinose can be used as osmotic agents to resist drought in plants [42]. The mesophyll tissue of cucumber seedlings under stress conditions showed the accumulation of raffinose, rather than stachyose, indicating that raffinose has a protective effect on cucumber seedlings under stress conditions. Whether *MnGolS2* plays a role through galactose, stachyose or raffinose under biotic stress remains to be studied.

Under stress conditions, plants switch between primary metabolism and secondary metabolism, and transfer available resources to defense against the outside world. Compared with wild poplars under salt stress, the overexpression of *AtGolS2* and *PtrGolS3* led to an increase in the content of soluble sugar in the two poplars with high GolS expression, and the content of other stress-related metabolites (proline, salicylic acid, phenylalanine) was also increased. This suggests that *PtrGolS* candidate genes can be used for glucose metabolic engineering to improve tolerance to abiotic stress [43]. Important secondary metabolites phenylalanine and tyrosine can also improve plant stress resistance [44]. Tricarboxylic acid cycle is an important primary metabolic pathway that generates energy and maintains the normal growth of plants. Several metabolites from the TCA cycle, such as oxaloacetate, malic acid, fumaric acid and citric acid, many hormones and other fatty acids derived from secondary metabolites, may be involved in stress, but the specific pathway remains to be explored.

Salicylic acid is an important plant hormone involved in plant biotic and abiotic stress responses [45,46,47]. Pathogenesis-related genes BL2 and PR1 may play a role in SA signaling pathway. Plants through activation of defense mechanisms in response to infection, including salicylic acid as a core signaling molecules, proteins associated with disease progression encoding genes related to the activation, the expression of which, PR-1, has been very widely used to establish SA-mediated defense and acquired resistance [48,49,50]. Under the condition of *B. cinerea* infection, the expression levels of BGL2 and PR 1 genes in *MnGolS2* transgenic plants were significantly higher than those in empty transgenic plants, indicating that they may through SA rely on signal transduction pathways involved in the resistance of *B. cinerea*. At the same time, SA mainly enhances the disease resistance of plants by properly regulating various enzyme activities and inducing the production of various reactive oxygen species [51,52]. Therefore, this study measured peroxidase (POD) and catalase (CAT). The results showed that the MDA content of *MnGolS2* transgenic plants decreased and CAT increased significantly, indicating that the activity of related enzymes in plants has been induced and involved in its resistance to gray mold. However, the specific pathway of MnGolS2 gene and the genes involved in the regulation of the expression of resistance to *B. cinerea* in this process still need to be further studied.

Studies have shown that galactinol and raffinose, in addition to their roles in carbon transport and storage, also have signaling mechanisms that promote ROS accumulation and protect plant cells from damage caused by ROS production, and galactinol synthase activity may be related to the concentration of RFO present in different plant organs. Just as RFO can induce tolerance to abiotic stresses such as dehydration, galactinol and RFO may also trigger signaling pathways that produce a defense response. The GolS gene was overexpressed, which can increase plant tolerance to different stresses [53]. *Arabidopsis* seeds specificity expressing *CaGolS1* and *CaGolS2*, induced by limiting the age excess ROS and subsequent lipid peroxidation, increase the seed vigor and life. In this paper, the functional mechanism of *MnGolS2* gene under biotic stress was studied for the first time, indicating that mulberry GolS gene has disease resistance. However, it is not clear how *MnGolS2* gene plays a role through SA signal transduction pathway. At the same time, studies have shown that the overexpression of *AtGolS2* and *PtrGolS3* is consistent with the small accumulation of transcripts of two genes involved in ABA signal regulation, which may increase the stress tolerance of plants [43]. Whether the upstream regulatory genes and their disease resistance are also related to the participation of other hormones remains to be further studied.

## 5. Conclusions

In the study, ectopic expression of the *MnGolS2* gene was confirmed by qRT-PCR and GUS staining. The *MnGolS2* transgenic plants inoculated with *B. cinerea* showed less leaf damage and produced less hydrogen peroxide and superoxide anion, suggesting that *MnGolS2* protects plant cells from the damage caused by ROS production, which indicated that they had disease resistance. At the same time, the expression levels of *AtBG2* and *AtPR1* in the *MnGolS2* transgenic plants were significantly increased under the infection of *B. cinerea*, and these two genes were involved in the salicylic acid signaling pathway, indicating that *MnGolS2* may resist the infection of *B. cinerea* through this pathway. It offers a crucial foundation for further research into the role of the GolS2 gene.

## Figures and Tables

**Figure 1 genes-14-01912-f001:**
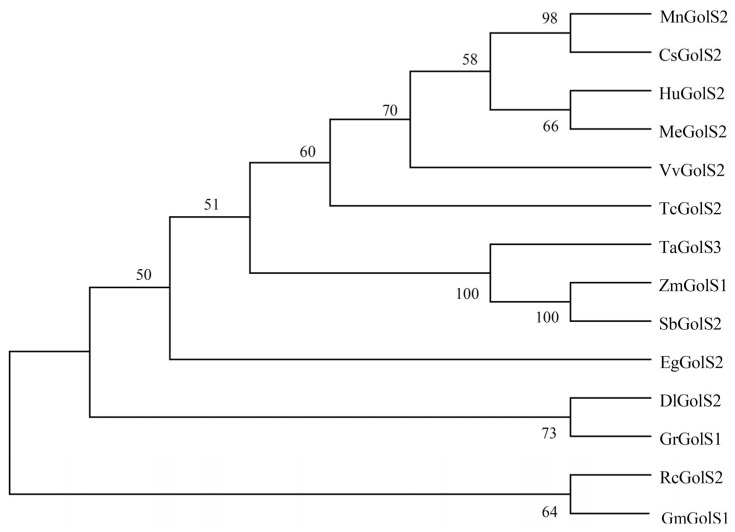
Phylogenetic tree of mulberry and other plants’ GolSs amino acid sequences. The neighbor-joining approach is used to construct the tree. A percentage is used to represent the bootstrap value, accession numbers obtained from GenBank are as follows: CsGolS2 (*Cannabis sativa*; XP_030504658.1), HuGolS2 (*Herrania umbratica*; XP_021282122.1), MeGolS2 (*Manihot esculenta*; XP_021623319.1), VvGolS2 (*Vitis vinifera*; XP_002281261.1), TcGolS2 (*Theobroma cacao*; XP_017974748.1), TaGolS3 (*Triticum aestivum*; XP_044322056.1), ZmGolS1 (*Zea mays*; NP_001105748.2), SbGolS2 (*Sorghum bicolor*; XP_002467954.1), EgGolS2 (*Eucalyptus grandis*; XP_010069751.2), *DlGolS2* (*Diospyros lotus*; XP_052184943.1), GrGolS1 (*Gossypium raimondii*; XP_012490292.1), RcGolS2 (*Rosa chinensis*; XP_024188412.1), GmGolS1 (*Glycine max*; XP_003554564.1).

**Figure 2 genes-14-01912-f002:**
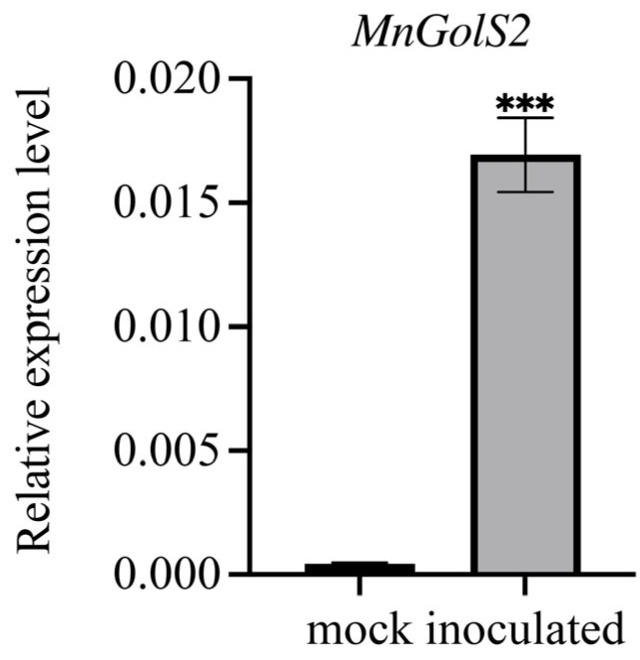
*MnGolS2* expression in mulberry leaves after mock-treated (Mock) and *B. cinerea* inoculation (Inoculated) is compared. The error bars show the standard deviation, and the sample size was three (*** *p*-value < 0.001; two-tailed *t*-test).

**Figure 3 genes-14-01912-f003:**
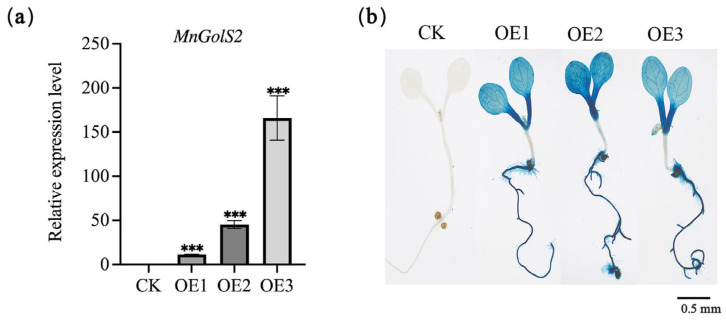
Identification of transgenic *Arabidopsis*. (**a**) Relative expression of *MnGolS2* in transgenic *Arabidopsis*. (**b**) GUS staining of transgenic *Arabidopsis*. CK, empty vector transgenic; OE, *MnGo1S2* transgenic. The error bars show the standard deviation, the sample size was three (*** *p*-value < 0.001; two-tailed *t*-test).

**Figure 4 genes-14-01912-f004:**
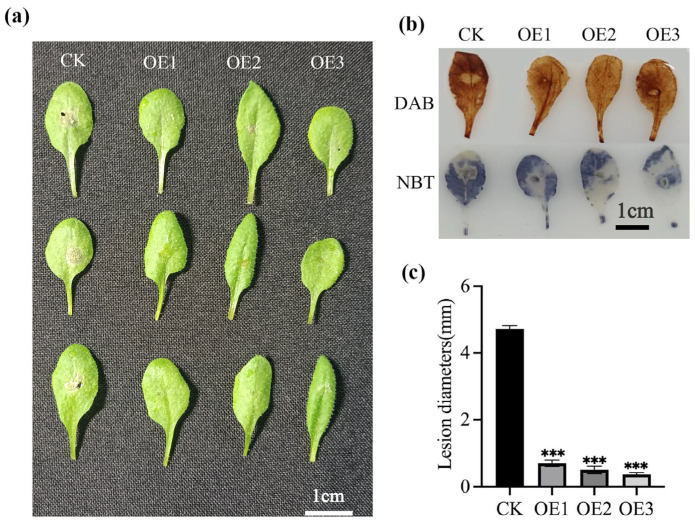
Resistance analysis of transgenic *Arabidopsis* to *B. cinerea*. (**a**) Leaf morphology of *Arabidopsis* 36 h after infection with *B. cinerea*. (**b**) DAB and NBT staining showed H_2_O_2_ and O^2−^ levels, respectively. (**c**) Quantitative analysis of transgenic *Arabidopsis* resistance. The error bar is standard deviation, the sample size was three (*** *p* < 0.001; two-tailed *t*-test).

**Figure 5 genes-14-01912-f005:**
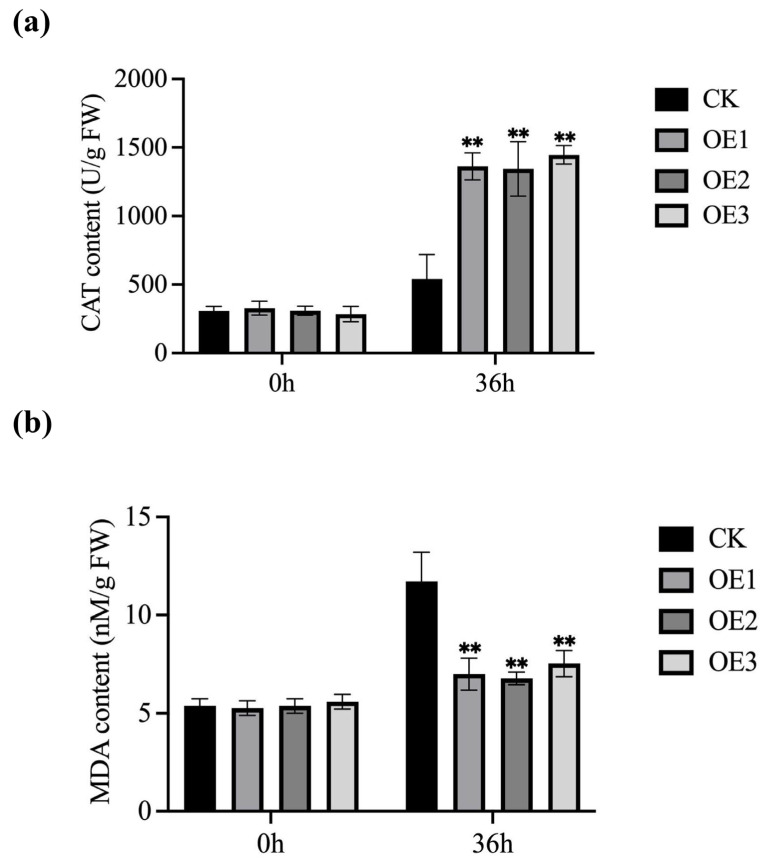
Physicochemical indicators were measured before and after *B. cinerea* inoculation. (**a**) The amount of malondialdehyde (MDA). (**b**) Activity of catalase (CAT). Error bars represent standard deviation; the sample size was three (** *p*-value < 0.01; two-tailed *t*-test).

**Figure 6 genes-14-01912-f006:**
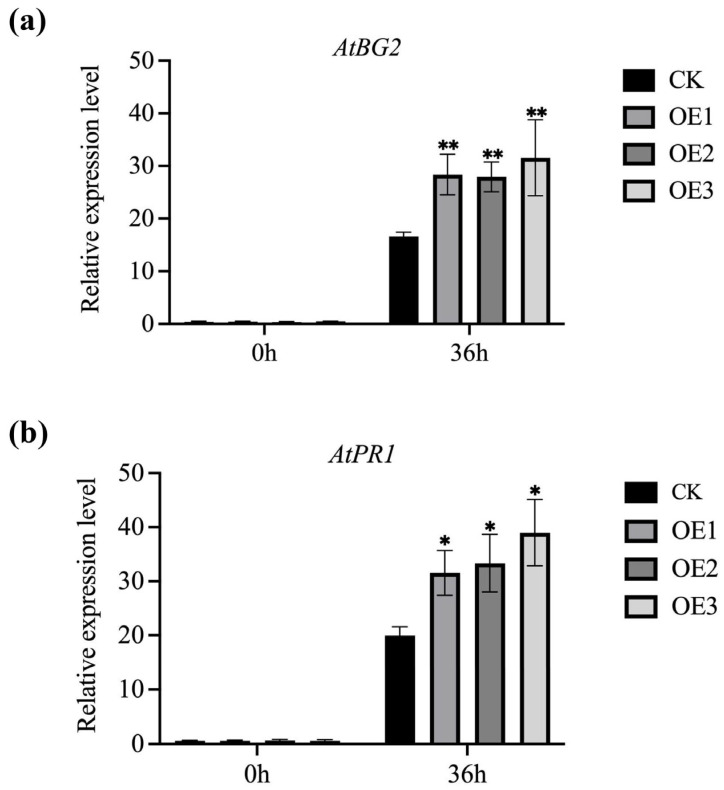
Relative expression of disease-course related genes in transgenic *Arabidopsis* before and after inoculation with *B. cinerea*. (**a**) *AtBG2* relative expression levels; (**b**) *AtPR1* relative expression levels. The error bar represents the standard deviation; the sample size was three (* *p*-value < 0.05, ** *p*-value < 0.01; two-tailed *t*-test).

## Data Availability

All data support the results of this study can be in online and its Appendix A obtained in the paper.

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
