# Peer review of "Mulberry *MnGolS2* Mediates Resistance to *Botrytis cinerea* on Transgenic Plants"

_genes, 2023, doi:10.3390/genes14101912_

Round 1

Reviewer 1 Report

The manuscript needs improvement mainly in the bibliography. In my view, authors should include in the introduction, main text, and discussion the suggested articles as presented below (1-8).

1. Martins, C.P., Fernandes, D., Guimarães, V.M., Du, D., Silva, D.D., Almeida, A.F., Gmitter, F.G., Otoni, W.C., & Costa, M.G. (2022). Comprehensive analysis of the GALACTINOL SYNTHASE (GolS) gene family in citrus and the function of CsGolS6 in stress tolerance. PLoS ONE, 17.

2.  Miyazawa, S., Nishiguchi, M., Kogawara, S., Tahara, K., Mohri, T., Kakegawa, K., Yokota, S., & Nanjo, T. (2017). Isolation of the drought- and salt-responsive galactinol synthase (GolS) gene from black poplar leaves and analysis of the transformants overexpressing GolS.

3. Zhou, J., Yang, Y., Yu, J., Wang, L., Yu, X., Ohtani, M., Kusano, M., Saito, K., Demura, T., & Zhuge, Q. (2013). Responses of Populus trichocarpa galactinol synthase genes to abiotic stresses. Journal of Plant Research, 127, 347 - 358.

4. You, J., Wang, Y., Zhang, Y., Dossa, K., Li, D., Zhou, R., Wang, L., & Zhang, X. (2018). Genome-wide identification and expression analyses of genes involved in raffinose accumulation in sesame. Scientific Reports, 8.

5. Yang, J., Ling, C., Liu, Y., Zhang, H., Hussain, Q., Lyu, S., Wang, S., & Liu, Y. (2022). Genome-Wide Expression Profiling Analysis of Kiwifruit GolS and RFS Genes and Identification of AcRFS4 Function in Raffinose Accumulation. International Journal of Molecular Sciences, 23.

6. Abkhoo, J., & Sabbagh, S.K. (2016). Control of Phytophthora melonis damping-off, induction of defense responses, and gene expression of cucumber treated with commercial extract from Ascophyllum nodosum. Journal of Applied Phycology, 28, 1333-1342.

7. Han, Q., Li, T., Zhang, L., Yan, J., Dirk, L.M., Downie, B., & Zhao, T. (2015). Functional Analysis of the 5′-Regulatory Region of the Maize ALKALINE ALPHA-GALACTOSIDASE1 Gene. Plant Molecular Biology Reporter, 33, 1361-1370.

8. Gu, L., Han, Z., Zhang, L., Downie, B., & Zhao, T. (2013). Functional analysis of the 5' regulatory region of the maize GALACTINOL SYNTHASE2 gene. Plant science : an international journal of experimental plant biology, 213, 38-45 .

Moreover, the bibliography needs to be explained and discussed better e.g., the possibility that galactinol and raffinose are related to oxidative damage was first discussed by Nishizawa et al., 2008, after by Sun et al., 2013 or by Sengupta et al., 2015 and later by Salvi et al (2016) as mentioned in the manuscript. Here the review manuscript by dos Santos and Vieira will help the readers understand the involvement of the galactinol synthase gene in abiotic and biotic stress responses.

Nishizawa, A., Yabuta, Y., & Shigeoka, S. (2008). Galactinol and Raffinose Constitute a Novel Function to Protect Plants from Oxidative Damage1[W][OA]. Plant Physiology, 147, 1251 - 1263.

Sun, Z., Qi, X., Wang, Z., Li, P., Wu, C., Zhang, H., & Zhao, Y. (2013). Overexpression of TsGOLS2, a galactinol synthase, in Arabidopsis thaliana enhances tolerance to high salinity and osmotic stresses. Plant physiology and biochemistry : PPB, 69, 82-9 .

Sengupta, S., Mukherjee, S., Basak, P., & Majumder, A.L. (2015). Significance of galactinol and raffinose family oligosaccharide synthesis in plants. Frontiers in Plant Science, 6.

Salvi, P., Saxena, S.C., Petla, B.P., Kamble, N.U., Kaur, H., Verma, P., Rao, V., Ghosh, S., & Majee, M. (2016). Differentially expressed galactinol synthase(s) in chickpea are implicated in seed vigor and longevity by limiting the age induced ROS accumulation. Scientific Reports, 6.

dos Santos, T.B., & Vieira, L.G. (2020). Involvement of the galactinol synthase gene in abiotic and biotic stress responses: A review on current knowledge. Plant Gene, 24, 100258.

Besides the above suggestions, the manuscript needs improvement to the reference list (numbering), depending on the content. The numbered list of sources (reference list) should be 1-43, not 1-20 (page 1) followed by 42 and 43 (page 3).

Author Response

Reviewer 1

The manuscript needs improvement mainly in the bibliography. In my view, authors should include in the introduction, main text, and discussion the suggested articles as presented below (1-8).

1.Martins, C.P., Fernandes, D., Guimarães, V.M., Du, D., Silva, D.D., Almeida, A.F., Gmitter, F.G., Otoni, W.C., & Costa, M.G. (2022). Comprehensive analysis of the GALACTINOL SYNTHASE (GolS) gene family in citrus and the function of CsGolS6 in stress tolerance. PLoS ONE, 17.

2.Miyazawa, S., Nishiguchi, M., Kogawara, S., Tahara, K., Mohri, T., Kakegawa, K., Yokota, S., & Nanjo, T. (2017). Isolation of the drought- and salt-responsive galactinol synthase (GolS) gene from black poplar leaves and analysis of the transformants overexpressing GolS.

3.Zhou, J., Yang, Y., Yu, J., Wang, L., Yu, X., Ohtani, M., Kusano, M., Saito, K., Demura, T., & Zhuge, Q. (2013). Responses of Populus trichocarpa galactinol synthase genes to abiotic stresses. Journal of Plant Research, 127, 347 - 358.

4.You, J., Wang, Y., Zhang, Y., Dossa, K., Li, D., Zhou, R., Wang, L., & Zhang, X. (2018). Genome-wide identification and expression analyses of genes involved in raffinose accumulation in sesame. Scientific Reports, 8.

5.Yang, J., Ling, C., Liu, Y., Zhang, H., Hussain, Q., Lyu, S., Wang, S., & Liu, Y. (2022). Genome-Wide Expression Profiling Analysis of Kiwifruit GolS and RFS Genes and Identification of AcRFS4 Function in Raffinose Accumulation. International Journal of Molecular Sciences, 23.

6.Abkhoo, J., & Sabbagh, S.K. (2016). Control of Phytophthora melonis damping-off, induction of defense responses, and gene expression of cucumber treated with commercial extract from Ascophyllum nodosum. Journal of Applied Phycology, 28, 1333-1342.

7.Han, Q., Li, T., Zhang, L., Yan, J., Dirk, L.M., Downie, B., & Zhao, T. (2015). Functional Analysis of the 5′-Regulatory Region of the Maize ALKALINE ALPHA-GALACTOSIDASE1 Gene. Plant Molecular Biology Reporter, 33, 1361-1370. 

8.Gu, L., Han, Z., Zhang, L., Downie, B., & Zhao, T. (2013). Functional analysis of the 5' regulatory region of the maize GALACTINOL SYNTHASE2 gene. Plant science : an international journal of experimental plant biology, 213, 38-45 .

Moreover, the bibliography needs to be explained and discussed better e.g., the possibility that galactinol and raffinose are related to oxidative damage was first discussed by Nishizawa et al., 2008, after by Sun et al., 2013 or by Sengupta et al., 2015 and later by Salvi et al (2016) as mentioned in the manuscript. Here the review manuscript by dos Santos and Vieira will help the readers understand the involvement of the galactinol synthase gene in abiotic and biotic stress responses.

Nishizawa, A., Yabuta, Y., & Shigeoka, S. (2008). Galactinol and Raffinose Constitute a Novel Function to Protect Plants from Oxidative Damage1[W][OA]. Plant Physiology, 147, 1251 - 1263.

Sun, Z., Qi, X., Wang, Z., Li, P., Wu, C., Zhang, H., & Zhao, Y. (2013). Overexpression of TsGOLS2, a galactinol synthase, in Arabidopsis thaliana enhances tolerance to high salinity and osmotic stresses. Plant physiology and biochemistry : PPB, 69, 82-9 .

Sengupta, S., Mukherjee, S., Basak, P., & Majumder, A.L. (2015). Significance of galactinol and raffinose family oligosaccharide synthesis in plants. Frontiers in Plant Science, 6.

Salvi, P., Saxena, S.C., Petla, B.P., Kamble, N.U., Kaur, H., Verma, P., Rao, V., Ghosh, S., & Majee, M. (2016). Differentially expressed galactinol synthase(s) in chickpea are implicated in seed vigor and longevity by limiting the age induced ROS accumulation. Scientific Reports, 6.

dos Santos, T.B., & Vieira, L.G. (2020). Involvement of the galactinol synthase gene in abiotic and biotic stress responses: A review on current knowledge. Plant Gene, 24, 100258.

Response 1: Based on your suggestions, the above papers have been cited as references 2, 4, 5, 6, 10, 12, 14, 15, 28, 37, 38, 41, 55.

Besides the above suggestions, the manuscript needs improvement to the reference list (numbering), depending on the content. The numbered list of sources (reference list) should be 1-43, not 1-20 (page 1) followed by 42 and 43 (page 3).

Response 2: According to your suggestions, we improve the 1-3 page reference list.

Reviewer 2 Report

Dear Authors, I have read the MS entitled Mulberry MnGolS2 mediates resistance to Botrytis cinerea on transgenic plants.

The article has as main aim analysing the influence of MnGolS2 over botrytis attacks in mulberry plants. The article has an interesting view over this subject, but there are some issues that must be corrected before being accepted for publication.

-why are there no numbers per line? is this according to journal requirements?

-please enrich the introduction with nore information of the importance of mulberry plants and also add bibliografical refferences (what studies are the authors reffering to here "Previous studies on GolS genes mainly focused on abiotic stress.....")

-please reorder the subchapters of Chapter 2 Material and Methods so that 2.1. clearly describes the plant material that was taken into study. Presently, this info is presented in an unclear and much too short manner in 2.3 (seedlings).

-please add a subchapter in Chapter 2 regarding origin of equipment and reactives used. 

-please check and correct if necessary : Karlsbad or Carlsbad?

-please check and correct if necessary: graphs in figure 2,3 and 6 have no measuring units on the OY axis?

-please add bibliographical refferences everywhere in text (mostly discussions chapter) where the words "Previous studies have shown" appear

-I suggest enriching the conclusions with facts extracted directly from the results of the experiments.

-please correct all over text "we" and active voice with passive voice

-please correct use of comma all over text. Usually there is no comma before "and"

Author Response

The article has as main aim analysing the influence of MnGolS2 over botrytis attacks in mulberry plants.The article has an interesting view over this subject, but there are some issues that must be corrected before being accepted for publication.

why are there no numbers per line? is this according to journal requirements?

Response 1:We used a template provided by the journal. There are no numbers in each line of the template.

please enrich the introduction with nore information of the importance of mulberry plants and also add bibliografical refferences (what studies are the authors reffering to here "Previous studies on GolS genes mainly focused on abiotic stress.....")

Response 2:According to your suggestion, we have added relevant information on the importance of mulberry plants in the Introduction, and added references to previous studies in the Introduction and discussion.

please reorder the subchapters of Chapter 2 Material and Methods so that 2.1. clearly describes the plant material that was taken into study. Presently, this info is presented in an unclear and much too short manner in 2.3 (seedlings).

Response 3:According to your suggestion, we have reordered the sections of materials and methods in Chapter 2, and added the plant materials and related equipment in 2.1. In 2.4, the research materials were supplemented in detail.

please add a subchapter in Chapter 2 regarding origin of equipment and reactives used.

Response 4:Based on your suggestions,we added plant materials and related equipment in 2.1.

please check and correct if necessary : Karlsbad or Carlsbad?

Response 5:According to your comments, it has been corrected to Carlsbad.

please check and correct if necessary: graphs in figure 2,3 and 6 have no measuring units on the OY axis?

Response 6:The figures in Figs.2, 3 and 6 are the results of Quantitative real-time PCR, the OY axis represents a relative expression without a unit of measurement.

please add bibliographical refferences everywhere in text (mostly discussions chapter) where the words "Previous studies have shown" appear

Response 7:According to your suggestion, we have added relevant references in the discussion section.

I suggest enriching the conclusions with facts extracted directly from the results of the experiments.

Response 8:According to your comments, we have modified the conclusions.

Comments on the Quality of English Language

-please correct all over text "we" and active voice with passive voice

-please correct use of comma all over text. Usually there is no comma before "and"

Response 9:According to your suggestion, we have corrected the whole text.

Reviewer 3 Report

Dear Author

The material and methods section is lacking missing details. Please find my comments below:

Page2  2.2. Quantitative real-time PCR:                                                             -This paragraph mentions isolating RNA from samples, What samples? Plant or fungi? How many samples? When and where was obtained?

Pag2  2.3. Construction of expression vector and plant transformation:            -Where the transgenics lines were made?

Page2  2.4. Analysis of resistance of transgenic Arabidopsis to B. cinerea:        -What pathogenicity protocol was followed?  Please mention it with details. 

-Need to mention the disease severity was assessed in the Material and Methods section.

-The protocol for NBT and DAB needs to be described in detail or cited in a paper.

- What is a DAB?

Author Response

Reviewer suggestion: The material and methods section is lacking missing details.

Page2 2.2. Quantitative real-time PCR This paragraph mentions isolating RNA from samples, What samples? Plant or fungi? How many samples? When and where was obtained?

Response 1: The RNA samples we isolated were mulberry and Arabidopsis thaliana leaves. One group of leaves was infected by Botrytis cinerea and the other group was not infected. The number of samples was three replicates, and the samples were obtained after 36 h of Botrytis cinerea infection. It is explained in material method 2.3、2.5.

Pag2  2.3. Construction of expression vector and plant transformation: Where the transgenics lines were made?

Response 2: We infected Arabidopsis thaliana by inflorescence infection method to obtain transgenic lines. This method has been reported by previous studies. The specific method can be seen in the reference 31 of this article.

Page2  2.4. Analysis of resistance of transgenic Arabidopsis to B. cinerea: What pathogenicity protocol was followed? Please mention it with details.

Need to mention the disease severity was assessed in the Material and Methods section.

Response 3: The resistance of transgenic Arabidopsis thaliana to botridomycetes was mainly assessed by the lesion area, which was measured by the LA-S type plant image analyzer, and was supplemented in Material methods 2.1 and 2.5 according to your suggestion

The protocol for NBT and DAB needs to be described in detail or cited in a paper.

What is a DAB?

Response 4: NBT is made from 0.1% nitroblue tetrazolium (NBT) containing 50 mM potassium hydrogen phosphate buffer (pH = 7.8), DAB is obtained by purchasing a kit produced by Solarbio. The study material is dyed in the above solution and then decolorized to obtain the desired result. Material method 2.5 is further explained.

3,3 ' -diaminobenzidine ( DAB ) is a chromogenic substrate of peroxidase. In the presence of hydrogen peroxide, DAB loses electrons and shows color change and accumulation, forming brown insoluble products, which are precipitated in situ.

Round 2

Reviewer 1 Report

The text and the references are arranged according to 1st round suggestions.

Author Response

The text and the references are arranged according to 1st round suggestions.

Response : According to your suggestions, we re-checked references 1-56 and typesetted them strictly in accordance with the requirements of the journal.

Reviewer 2 Report

Dear Authors, thank you for revising and resubmitting your research. The article has been much more improved. However, there still are some minor format issues that should be resolved.

-usually, the title of the figure should be on the same page as the figure itself (see fig. 1)

-the measuring unit is placed after a space (19 oC). Please correct all over text.

-i have suggested enriching the conlsuions with data extracted from the results and discussions part. Now this has become even shorter?

-as mentioned before, scientific materials should not use active voice, but passive voice. Therefore, the use of "we" should be discarded and "the authors" or "the research" should be used instead. Please correct all over text.

Author Response

-usually, the title of the figure should be on the same page as the figure itself (see fig. 1)

Response 1: According to your suggestion, we have placed the title of the figure on the same page as the figure itself and strictly followed the typographical requirements of the journal.

-the measuring unit is placed after a space (19 °C ). Please correct all over text.

Response 2: According to your suggestion, we have revised the full text.

-i have suggested enriching the conlsuions with data extracted from the results and discussions part. Now this has become even shorter

Response 3: Based on your suggestions, we have supplemented and enriched the results. the conclusionsas mentioned before, scientific materials should not use active voice, but passive voice. Therefore, the use of "we" should be discarded and "the authors" or "the research" should be used instead. Please correct all over text.

Response 3: According to your suggestion, we changed the conclusion and other parts into the passive voice and abandoned the use of "we"

Reviewer 3 Report

Dear Author:

The material and methods need to be clear with details. Are lot of info is still missing:

Material and methods section: 2.1. Plant materials and equipment:

This is the 1st step that all the unscripted results depend on, if is not done right the result will not be robust. If the isolate used for the test is not a pathogenes that will trigger a different expression gene. 

-What is the plant inoculated with? 

-From where was the isolate obtained from?

-What is the isolate number?

-How the pathogen was grown?

-Is the pathogenisity of the isolate already tested? 

-What was the spore suspension that was used for inoculation?

-What is the model, year, and company of the growth chamber used in this experiment?

Author Response

This is the 1st step that all the unscripted results depend on, if is not done right the result will not be robust. If the isolate used for the test is not a pathogenes that will trigger a different expression gene.

-What is the plant inoculated with?

Response 1: First, we activated B. cinerea,then, an inoculation was carried out with an Agar block containing B. cinerea mycelium.

-From where was the isolate obtained from?

Response 2: We used a previously preserved B. cinerea strain that had been previously identified and determined to be pathogenic to the flowers, fruits and leaves of the mulberry tree.

-What is the isolate number?

Response 3: The isolate is numbered MM1 isolate. It is also mentioned in reference 27.

-How the pathogen was grown?

Response 4: B. cinerea was cultured on Potato Dextrose Agar at 28 °C.

-Is the pathogenisity of the isolate already tested?

Response 5: Previously, we have isolated and identified B. cinerea, it is confirmed that it is indeed pathogenic to the flowers, leaves, fruits and other organs of mulberry. In addition, we have published relevant papers , as follows:

Amplification of miniature inverted-repeat transposable elements and the associated impact on gene regulation and alternative splicing in mulberry (Morus notabilis).

Mulberry genes MnANR and MnLAR confer transgenic plants with resistance to Botrytis cinerea.

Dynamic mulberry (Morus notabilis) transposable element and gene methylation changes in response to Botrytis cinerea.

Characterization of the Chitinase Gene Family in Mulberry (Morus notabilis) and MnChi18 Involved in Resistance to Botrytis cinerea.

-What was the spore suspension that was used for inoculation?

Response 6: Instead of inoculating with spore suspensions, Potato Dextrose was used Agar for culture inoculation.

-What is the model, year, and company of the growth chamber used in this experiment?

Response 7: We use the intelligent light incubator of Jiangnan Ningbo Instrument Factory, the model is GXZ, the year is 2020.Relevant content has been added in 2.1.